# Human Bone Marrow Mesenchymal Stromal/Stem Cells Regulate the Proinflammatory Response of Monocytes and Myeloid Dendritic Cells from Patients with Rheumatoid Arthritis

**DOI:** 10.3390/pharmaceutics14020404

**Published:** 2022-02-12

**Authors:** Paula Laranjeira, Mónia Pedrosa, Cátia Duarte, Susana Pedreiro, Brígida Antunes, Tânia Ribeiro, Francisco dos Santos, António Martinho, Margarida Fardilha, M. Rosário Domingues, Manuel Abecasis, José António Pereira da Silva, Artur Paiva

**Affiliations:** 1Flow Cytometry Unit, Department of Clinical Pathology, Centro Hospitalar e Universitário de Coimbra, Av. Bissaya Barreto, Bloco de Celas, 3000-075 Coimbra, Portugal; 1979paula@gmail.com; 2Centro do Sangue e da Transplantação de Coimbra, Instituto Português do Sangue e da Transplantação, Coimbra, Portugal, Quinta da Vinha Moura, São Martinho do Bispo, 3041-861 Coimbra, Portugal; pedrosa.monia@gmail.com (M.P.); susanapedreiro@gmail.com (S.P.); antonio.martinho@ipst.min-saude.pt (A.M.); 3Coimbra Institute for Clinical and Biomedical Research (iCBR), Faculty of Medicine, University of Coimbra, Pólo das Ciências da Saúde, Azinhaga de Santa Comba, 3000-548 Coimbra, Portugal; catiacmduarte@gmail.com (C.D.); jdasilva@ci.uc.pt (J.A.P.d.S.); 4Center for Innovative Biomedicine and Biotechnology (CIBB), University of Coimbra, Pólo das Ciências da Saúde, Azinhaga de Santa Comba, 3000-548 Coimbra, Portugal; 5Center for Neuroscience and Cell Biology (CNC), University of Coimbra, Polo 1, 1.° Piso, FMUC, Rua Larga, 3004-504 Coimbra, Portugal; 6Signal Transduction Laboratory, Center of Cellular Biology, SACS and Department of Biology, University of Aveiro, Campus Universitário de Santiago, 3810-193 Aveiro, Portugal; mfardilha@ua.pt; 7Enzifarma—Diagnostica e Farmacêutica, S.A., Estrada da Luz, n.° 90, 2° F, 1600-160 Lisbon, Portugal; 8Rheumatology Department, Hospitais da Universidade de Coimbra, Centro Hospitalar e Universitário de Coimbra, Praceta Professor Mota Pinto, 3000-075 Coimbra, Portugal; 9Cell2B Advanced Therapeutics, SA, Biocant Park, Núcleo 04, Lote 4A, 3060-197 Cantanhede, Portugal; brigidantunes@hotmail.com; (B.A.); catarino.tribeiro@gmail.com (T.R.); francisco.santos@crioestaminal.pt (F.d.S.); 10Stemlab SA, Biocant Park, Núcleo 04, Lote 2, 3060-197 Cantanhede, Portugal; 11Laboratory of Signal Transduction, Institute of Biomedicine—iBiMED, Department of Medical Sciences, University of Aveiro, Campus Universitário de Santiago, 3810-193 Aveiro, Portugal; 12Mass Spectrometry Centre, LAQV REQUIMTE, Department of Chemistry, University of Aveiro, Santiago University Campus, 3810-193 Aveiro, Portugal; mrd@ua.pt; 13Centre for Environmental and Marine Studies (CESAM), Department of Chemistry, University of Aveiro, Santiago University Campus, 3810-193 Aveiro, Portugal; 14Serviço de Transplantação de Progenitores Hematopoiéticos (UTM), Instituto Português de Oncologia de Lisboa Francisco Gentil, Rua Professor Lima Basto, 1099-023 Lisbon, Portugal; manuel.abecasis@ipst.min-saude.pt; 15Instituto Português do Sangue e da Transplantação—CEDACE, Alameda das Linhas de Torres, 117, 1769-001 Lisbon, Portugal; 16Instituto Politécnico de Coimbra, ESTESC-Coimbra Health School, Ciências Biomédicas Laboratoriais, Rua 5 de Outubro, 3046-854 Coimbra, Portugal

**Keywords:** mesenchymal stromal cells, mesenchymal stem cells, immunomodulation, rheumatoid arthritis, dendritic cells, monocytes, cytokines, chemokines

## Abstract

Rheumatoid arthritis (RA) is a disabling autoimmune disease whose treatment is ineffective for one-third of patients. Thus, the immunomodulatory potential of mesenchymal stromal/stem cells (MSCs) makes MSC-based therapy a promising approach to RA. This study aimed to explore the immunomodulatory action of human bone marrow (BM)-MSCs on myeloid dendritic cells (mDCs) and monocytes, especially on cytokines/chemokines involved in RA physiopathology. For that, LPS plus IFNγ-stimulated peripheral blood mononuclear cells from RA patients (n = 12) and healthy individuals (n = 6) were co-cultured with allogeneic BM-MSCs. TNF-α, CD83, CCR7 and MIP-1β protein levels were assessed in mDCs, classical, intermediate, and non-classical monocytes. mRNA expression of other cytokines/chemokines was also evaluated. BM-MSCs effectively reduced TNF-α, CD83, CCR7 and MIP-1β protein levels in mDCs and all monocyte subsets, in RA patients. The inhibition of TNF-α production was mainly achieved by the reduction of the percentage of cellsproducing this cytokine. BM-MSCs exhibited a remarkable suppressive action over antigen-presenting cells from RA patients, potentially affecting their ability to stimulate the immune adaptive response at different levels, by hampering their migration to the lymph node and the production of proinflammatory cytokines and chemokines. Accordingly, MSC-based therapies can be a valuable approach for RA treatment, especially for non-responder patients.

## 1. Introduction

Rheumatoid arthritis (RA) is an autoimmune disease, associated to Th1/Th17-mediated chronic inflammation of the joints, whose etiology is still elusive. Symmetric polyarthritis, affecting especially the hands and feet, is a hallmark of this disease [1,2,3]. Several types of immune cells, namely monocytes/macrophages and dendritic cells (DCs), actively participate in RA pathophysiology and, together with Th1 and Th17 cells, infiltrate the RA joint. There, they produce inflammatory mediators—namely interleukin (IL)-1, IL-6, tumor necrosis factor (TNF)-α, extracellular matrix-degrading enzymes, and free radicals—leading to chronic joint inflammation, with consequent cartilage destruction and bone erosion [1,3,4,5,6].

The introduction of disease-modifying anti-rheumatic drugs (DMARDs), namely biological agents targeting cytokines implicated in RA physiopathology, had definitely changed the clinical course of RA. Notwithstanding, DMARDs are not an effective treatment for all patients [2] and have been associated to an augmented risk of infections [7,8], which constitutes an important limitation. In this scenario, mesenchymal stromal/stem cells (MSCs) emerge as an alternative treatment for RA. A recent clinical trial in RA, reported that the administration of umbilical cord blood MSCs resulted in the reduction DAS28 score and peripheral inflammatory parameters, including TNF-α, IL-1β, IL-6 and IL-8, accompanied by the increase of IL-10 expression. These data suggest MSC therapy influences the course of RA with evident clinical improvement [9]. Encouraging results had also been described in other autoimmune diseases, like systemic lupus erythematosus [10,11,12] and multiple sclerosis [13,14]. Notwithstanding, clinical trials point out that multiple MSCs infusions at different time points will be probably needed and, in this scenario, immune sensitization against allogeneic MSCs may be a limitation for their use [15,16].

In the preclinical setting, MSC treatment of mice with collagen-induced arthritis (CIA) delayed the disease onset and abolished arthritis progression. Mice infused with MSCs also displayed decreased paw swelling, decreased immune infiltrate into the joints, and reduction of proinflammatory cytokines levels, along with increased IL-10 expression and regulatory T cells percentage [5,17,18].

Our research group had previously reported that human bone marrow (BM)-derived MSCs, co-cultured with peripheral blood mononuclear cells (PBMCs) from healthy individuals, resulted in the inhibition of inflammatory mediators production by myeloid dendritic cells (mDCs) and all the monocyte subpopulations identified in the peripheral blood (PB) (classical, intermediate and non-classical monocytes) [19], as well as different CD4^+^ and CD8^+^ T cell subsets, namely Th1, Th17, and Th9 [20]. Recently, we demonstrated MSC-mediated immunomodulation was maintained for T cell subsets from RA patients [21].

Here, we explore if this regulatory action is simultaneously exerted on RA antigen-presenting cells (APCs). For that, PBMCs (from the same RA patients and control group enrolled in our former study [21]) were cultured alone or in the presence of allogeneic BM-MSCs, in order to investigate the influence of BM-MSCs on the protein levels of TNF-α, CD83, CCR7, and CCL4 (or macrophage inflammatory protein, MIP-1β) by mDCs and monocyte subpopulations (classical, intermediate, and non-classical monocytes). Furthermore, mRNA levels of IL-1β, CXCL9, CXCL10, CCL3 and CCL5, were measured in fluorescence-activated cell sorting (FACS)-purified mDCs, classical, and non-classical monocytes.

This study demonstrated that BM-MSCs exert a significant inhibitory action over all monocyte subsets and mDCs from RA patients, potentially affecting their ability to stimulate the immune adaptive response at different levels: BM-MSCs hinder CCR7 upregulation upon cell activation, and this can potentially hamper APCs’ migration into the lymph node; simultaneously, BM-MSCs inhibit the proinflammatory cytokines/chemokines’ production by mDCs and monocytes, which can potentially hamper T cell polarization towards Th1/Th17 and migration to inflamed tissue. Accordingly, MSC therapies can be valuable for RA treatment, especially for non-responder patients.

## 2. Materials and Methods

### 2.1. Collection of Peripheral Blood and Gradient Density Separation of PBMCs

The collection of PB samples in heparin was carried out at the Centro Hospitalar e Universitário de Coimbra (CHUC, Rheumatology Unit), Portugal, and Instituto Português do Sangue e da Transplantação (Centro do Sangue e da Transplantação de Coimbra), Portugal. The study enrolled six healthy donors (five females and one male; mean age: 44 ± 7 years, range: 22–51 years old) and 12 RA patients (eight females and four males; mean age: 53 ± 9 years, range: 38–71 years old).

Five patients were classified as patients with inactive RA (DAS28-CRP3v = 1.9 ± 0.8) and seven as patients with active disease (DAS28-CRP3v = 4.6 ± 0.7), according to the disease activity score 28 using CRP level (DAS28-CRP; 3-variable). Diagnosis of rheumatoid arthritis was made in accordance with the American College of Rheumatology 1987 Criteria or ACR/EULAR criteria 2010. Clinical and demographic data about the individuals enrolled in this study are detailed in Appendix A
Table A1. RA patients treated with biologics (such as rituximab, tocilizumab, or anti-TNF), with other autoimmune or inflammatory diseases, previous cancer, infection or other acute or chronic diseases, were excluded from this study. PBMCs isolation were performed by gradient density centrifugation, using Lymphoprep (Stemcell Technologies, Vancouver, BC, Canada), and by centrifuging at 800× *g* for 20 min. Then, HBSS (Gibco, Life Technologies, Paisley, UK) was used to wash PBMCs, which were finally resuspended in RPMI 1640 medium supplemented with GlutaMax (Invitrogen, Life Technologies, Waltham, MA, USA) and antibiotic-antimycotic (Gibco, at a final concentration of 100 units/mL of penicillin, 100 µg/mL of streptomycin, and 0.25 µg/mL of Gibco amphotericin B).

The analysis of protein and mRNA expression by mDCs and monocytes was carried out in PBMCs cultured with or without allogeneic BM-MSCs, in the presence/absence of the stimulation agents lipopolysaccharide (LPS) and interferon (IFN)γ, as follows: (1) 10^6^ PBMCs + 500 μL RPMI (negative control); (2) 10^6^ PBMCs + 0.5 × 10^6^ MSCs (negative control); (3) 10^6^ PBMCs + LPS + IFNγ (positive control); (4) 10^6^ PBMCs + 0.5 × 10^6^ MSCs + LPS + IFNγ; (5) 10^6^ PBMCs + 0.5 × 10^6^ MSCs, with subsequent BM-MSCs depletion and, after that, LPS plus IFNγ stimulation. The experimental protocols used here had been described in a previous work from our group [19] and are detailed in the following sections.

### 2.2. Isolation of Human BM-MSCs

BM-MSCs were isolated from eight healthy BM donors admitted to the Instituto Português de Oncologia de Lisboa Francisco Gentil (Serviço de Transplantação de Progenitores Hematopoiéticos, UTM), Portugal.. Sepax S-100 system (Biosafe, Eysins, Switzerland) was used to isolate PBMCs from BM samples, by following the manufacturer’s instructions. Trypan Blue (Gibco) exclusion method were performed to determine cell count and cell viability. BM PBMCs, plated in 10% qualified fetal bovine serum (FBS, Sigma, Madrid, Spain) supplemented DMEM (Gibco), at a 2 × 10^5^ cells/cm^2^ density, were incubated for 3 days, at 37 °C, in 5% CO_2_ sterile and humidified atmosphere. After discarding the non-adherent cell fraction, the adherent cells were maintained in culture with a complete medium renewal every 3 to 4 days, until reach a 70–80% confluency. At that point, a 7 min incubation with TrypLE (Life Technologies) was performed to detach the cells, following their replating at a density of 3000 cells/cm^2^. BM-MSCs between passage 3 and 5 were used in the present study, and their identity was confirmed by immunophenotype characterization, fluorescent morphological analysis, and osteogenic, adipogenic, and chondrogenic mesodermal differentiation assays, in accordance to the Mesenchymal and Tissue Stem Cell Committee of the International Society for Cellular Therapy [22]. Protein levels of CD271 were also evaluated by flow cytometry. The Ethics Committee of Centro Hospitalar e Universitário de Coimbra approved this study (CHUC-086-16), and written informed consent were obtained from all participants.

### 2.3. Co-Culture of PBMCs and BM-MSCs

The co-culture system used here is described in a former study from our group [19]: 10^6^ PBMCs alone, or 10^6^ PBMCs + 0.5 × 10^6^ allogeneic MSCs (ratio PBMCs:MSCs = 2:1) were placed in tissue culture plates (Falcon, Becton Dickinson Biosciences (BD), San Jose, CA, USA) in a total volume of 1 mL of RPMI 1640 medium supplemented with GlutaMax (Invitrogen) and antibiotic/antimycotic (Gibco, at a final concentration of 100 units/mL of penicillin, 100 µg/mL of streptomycin, and 0.25 µg/mL of Gibco amphotericin B). After 20 h of incubation at 37 °C, in humidified and sterile atmosphere, containing 5% CO_2_, we proceeded to MSCs’ depletion, in part of PBMCs + MSCs co-cultures, with the EasySep Human CD271 Selection kit (Stemcell Technologies, Vancouver, BC, Canada), following the instructions of the manufacturer.

Lipopolysaccharide (LPS, 100 ng/mL) and interferon (IFN)γ (100 U/mL) were used to stimulate PBMCs. In addition, 10 μg/mL of brefeldin A, from *Penicillium brefeldianum* (Sigma), was added to cell cultures devoted to the study of TNF-α and MIP-1β protein production, by flow cytometry. Brefeldin A blocks protein transport to the Golgi complex, leading to the accumulation of proteins in the endoplasmic reticulum, whose levels can be assessed by flow cytometry, using an intracellular staining protocol. No brefeldin A was added to the cell cultures that would be used for the evaluation of CD83 and CCR7 protein levels (using flow cytometry), nor for mRNA levels of cytokines, in purified monocytes and mDCs. This was followed by 6 h of incubation in the same conditions. In sum, the study of the proteins’ and mRNA expression was systematically performed in each one of the following culture conditions: (1) PBMCs; (2) PBMCs + MSCs; (3) PBMCs + LPS + IFNγ; (4) PBMCs + MSCs + LPS + IFNγ; (5) PBMCs + MSCs + MSCs’ depletion + LPS + IFNγ.

### 2.4. Immunophenotypic Study of Monocyte Subsets and mDCs

#### 2.4.1. Staining Protocol

As described previously by our group [19], a 10 min incubation with TrypLE (Gibco) was performed to detached cells from tissue culture plates, which were then transferred to a 12 mm × 75 mm cytometer tube, and centrifuged at 540× *g* for 5 min. The supernatant was discarded. An eight-color monoclonal antibody (mAb) combinations’ panel was used to the phenotypic study of PB monocytes and mDCs (Table 1). To study CD83 and CCR7 protein levels (Table 1, tube 1), a stain-lyse-and-then-wash protocol was used [19]: cells were stained with mAbs and incubated for 10 min, in the dark, at room temperature; then 2 mL of FACSLysing Solution (BD) was added, followed by a 10 min period of incubation; finally, cells were washed with 1 mL of PBS (540× *g*, 5 min), the cell pellet was resuspended in 500 μL and immediately acquired in a FACSCanto II (BD) flow cytometer. To evaluate TNF-α and MIP-1β protein production (Table 1, tube 2), cells were stained for surface protein antigens, in a first step, followed by a 10 min incubation in the dark, at room temperature; after that, cells were washed with PBS; and then stained for the intracellular protein antigens, using Fix&Perm (Caltag, Hamburg, Germany), and following the manufacturer’s instructions [19]. After incubating for 15 min, in the dark, at room temperature, with the mAbs against the intracellular antigens, cells were washed twice with 1 mL of PBS (540× *g*, 5 min), the cell pellet was resuspended in 500 μL of PBS and immediately acquired in a FACSCanto II (BD) flow cytometer.

#### 2.4.2. Data Acquisition and Analysis

Data were acquired in a FACSCanto II (BD) flow cytometer equipped with the FACSDiva software (v6.1.2; BD, San Jose, CA, USA). For all samples, the number of events stored was always above 0.5 × 10^6^. Data analysis was performed using the Infinicyt software (version 1.7; Cytognos SL, Salamanca, Spain).

#### 2.4.3. Identification and Characterization of PB mDCs and Classical, Intermediate, and Non-Classical Monocytes, by Flow Cytometry

For the identification of classical, intermediate, and non-classical monocytes, and mDCs, we applied the gating strategy reported in our previous study [19] and depicted in Figure 1. In short, classical monocytes display high levels of CD14, HLA-DR and CD33, are positive for IREM-2 (CD300e), and negative for CD16; intermediate monocytes are positive for CD14, with an increasing CD16 levels, and lower levels of CD33, in comparison to classical monocytes; lastly, non-classical monocytes are positive for CD16, displaying low to negative reactivity for CD14, and presenting the lowest CD33 levels and the highest CD45 levels, amongst the three PB monocyte subpopulations; in turn, mDCs have lower side-scatter (SSC) and lower levels of CD45, compared to monocytes, along with high levels of HLA-DR and CD33, and are negative for CD14, CD300e and CD16.

The strategy used to identify monocytes and mDCs producing TNF-α and MIP-1β is illustrated in Figure 2. To evaluate the levels of these two proteins by flow cytometry, brefeldin A was added to the cell cultures in order to block the transport of the newly synthesized proteins at intracellular level. Then, an intracellular staining protocol was used to evaluate, at intracellular level, and within each one of the immune cell populations under study, the protein levels of TNF-α and MIP-1β.

### 2.5. Cell Purification of Classical Monocytes, Non-Classical Monocytes and mDCs

A FACSAria II flow cytometer (BD) was used to purify classical monocytes, non-classical monocytes and mDCs. The combination of mAbs described in Table 1 (tube 3) enabled classical monocytes (CD14^++^ CD16^−^ HLA-DR^+^ CD33^+^ CD300e^+^), non-classical monocytes (CD14^dim/−^ CD16^+^ HLA-DR^+^ CD33^+^ CD300e^+^), and mDCs (CD14^−^ CD16^−^ HLA-DR^++^ CD33^++^ CD300e^−^) identification. Studies of mRNA expression were subsequently performed in the purified cell subsets.

### 2.6. mRNA Expression in the Purified Cell Populations

The experimental protocol for the study of mRNA expression by purified classical monocytes, non-classical monocytes and mDCs had been previously described by our group [19]. Purified cells were resuspended in RLT Lysis Buffer (Qiagen, Hilden, Germany). Total RNA were extracted, using RNeasy Micro kit (Qiagen), as per manufacturer recommendations, and eluted in RNase-free water, in a final volume of 20 µL. After reverse transcription with Tetra cDNA Synthesis^®^ (Bioline, London, UK) we preformed real-time (RT) PCR, using the LightCycler 480 II (Roche Diagnostics, Rotkreuz, Switzerland), for the relative quantification of gene expression. RT-PCR reactions were performed with QuantiTect SYBR Green PCR Master Mix (Qiagen), and QuantiTect Primer Assay (CXCL9: QT00013461; CXCL10: QT01003065; CCL3: QT01008063; CCL5: QT00090083; IL-1β: QT00021385) (Qiagen), in a final volume of 10 μL, and all samples were run in duplicate. We used the thermal profile previously described by our group [19] for the polymerase chain reactions: 95 °C for 10 min (1 cycle), then 50 cycles of 95 °C for 10 s, 55 °C for 20 s, 72 °C for 30 s, 1 cycle of 95 °C for 5 s, 65 °C for 60 s, and continuous at 97 °C; at last, 1 cycle of 21 °C for 10 s. The analysis of RT-PCR results was performed using the LightCycler software (Roche Diagnostics). Reference genes selection and data normalization were performed in GeNorm software (PrimerDesign Ltd., Southampton, UK). As reference genes for classical monocytes, we selected topoisomerase DNA I (TOP1) and glyceraldehyde-3-phosphate dehydrogenase (GAPDH); while the reference genes for non-classical monocytes and mDCs were β-2 microglobulin (B2M) and GAPDH. The delta-Ct method was used to calculate the normalized expression levels of the genes of interest. The mRNA expression of CCL3, CCL5, CXCL9, CXCL10 and IL-1β was determined in purified classical monocytes and non-classical monocytes, whereas CXCL10 and IL-1β mRNA expression was measured in purified mDCs.

### 2.7. Statistical Analyses

Data were presented as the mean values ± standard deviation. The Wilcoxon, Friedman, and Mann-Whitney non-parametric tests were applied to determine the significance of the differences between the different experimental conditions, as appropriate, with the Statistical Package for Social Sciences (IBM SPSS, version 17.0, Armonk, NY, USA) software. Differences were considered statistically significant when *p* < 0.05.

## 3. Results

To investigate how allogeneic BM-MSCs regulate the immune function of PB monocytes and mDCs from RA patients, we evaluated the protein levels or mRNA expression of proinflammatory cytokines (TNF-α and IL-1β), proteins involved in cell migration (CCL3 or MIP-1α, CCL4 or MIP-1β, CCL5 or RANTES, CXCL9 or MIG, CXCL10 or IP-10, and CCR7), and the maturation marker CD83, in the presence/absence of BM-MSCs and stimulating agents (LPS plus IFNγ). RA patients with inactive and active disease were considered together because the immunomodulatory behavior of BM-MSCs was similar for both groups of patients, as statistically tested. Likewise, no differences were found when comparing female vs. male RA patients, therefore, they were included in the same group. More detailed data, discriminating inactive and active RA patients, can be found in Appendix A
Table A2; and data on female vs. male patients in Appendix A
Figure A1.

### 3.1. BM-MSCs Hamper the Production of TNF-α and MIP-1β by Monocytes and mDCs from RA Patients and Healthy Individuals

In order to assess the effect of BM-MSCs over TNFα and MIP-1β protein production, we used flow cytometry to measure the intracellular levels of these cytokines in monocytes and mDCs, using PBMCs cultured in the absence or in the presence of BM-MSCs.

The presence of BM-MSCs in the cell culture resulted in an inhibitory effect over TNFα and MIP-1β protein production, transversal to all monocyte subsets (*p* < 0.05) and mDCs (*p* < 0.05), from both RA patients and healthy group (HG), as illustrated in Figure 3, Figure 4 and Figure 5. For TNFα, this inhibitory effect was mainly caused by the decreased percentage of cells producing cytokines (Figure 3); while for MIP-1β, it was achieved not only by the decrement of the percentage of MIP-1β^+^ cells, but also by the reduction of the amount of cytokine produced per cell, quantified as mean fluorescence intensity (MFI), as shown in Figure 4. Interestingly, the inhibition of MIP-1β was more pronounced in non-classical monocytes than in the remaining monocyte subpopulations. Concerning TNF-α, the inhibitory effect of BM-MSCs were stronger over mDCs than monocytes (Figure 5). Interestingly, the inhibitory effect of BM-MSCs was verified even in the assays where BM-MSCs were depleted immediately before PBMCs’ stimulation. Of note, BM-MSCs depletion prior to cell stimulation partially restored TNF-α production by classical and intermediate monocytes, but only from HG (*p* < 0.05).

Finally, we observed that unstimulated monocytes and mDCs from RA patients showed a basal production of TNF-α and MIP-1β, not detected among the HG (*p* < 0.05). On the other hand, monocytes from HG displayed a higher response to cell stimulation than those from RA patients (*p* < 0.05), (Figure 3, Figure 4 and Figure 5).

### 3.2. Effect of BM-MSCs on mRNA Levels of CXCL9, CXCL10, CCL3, CCL5, and IL-1β, in Monocytes and mDCs from RA Patients and Healthy Individuals

In the same line, LPS plus IFNγ stimulation of PBMCs from HG showed a tendency to increase mRNA levels of all cytokines/chemokines under study, which was not verified for RA patients; as result, in RA patients, the cytokine’s mRNA expression upon cell activation was significantly lower as compared to HG (*p* < 0.05), except for IL-1β in non-classical monocytes (Figure 6). As observed for TNF-α and MIP-1β, at protein level, a prior contact of 20 h of PBMCs with BM-MSCs sufficed for an inhibitory effect over mRNA cytokine and chemokine expression, not being necessary the presence of BM-MSCs during the PBMCs activation period.

Overall, our results suggest that BM-MSCs have propensity to inhibit the expression of the analyzed cytokines/chemokines at mRNA level in HG (with no statistical significance); while in RA patients this inhibitory trend is only verified for IL-1β in classical (*p* < 0.05) and non-classical monocytes, and for CCL3 in non-classical monocytes (*p* > 0.05), as showed in Figure 6.

### 3.3. BM-MSCs Reduce the Percentage of CCR7^+^ and CD83^+^ Monocytes and mDCs in RA Patients

Under our experimental settings, the protein levels of CCR7 and CD83 were upregulated by both BM-MSCs or LPS plus IFNγ stimulation; also, CCR7 induction upon cell stimulation was stronger for RA in comparison to HG (*p* < 0.05). It is interesting to notice that, in the HG, the intermediate monocytes stand out in relation to the remaining monocytes by displaying the highest upregulation of CCR7, upon LPS plus IFNγ stimulation. In turn, in RA patients, LPS plus IFNγ stimulation resulted in similar percentages of CCR7^+^ cells for all monocyte subpopulations, which were higher than those observed for HG. Besides, a marked difference, among HG and RA patients, was verified for the percentage of CCR7^+^ cells in classical and non-classical monocytes (Figure 7).

Interestingly, a suppressive effect of BM-MSCs on CCR7 and CD83 induction was detected in activated monocytes (classical, intermediate and non-classical) and mDCs from RA patients, whereas no inhibitory effect was found in HG (Figure 7 and Figure 8). Once again, this inhibitory effect was maintained even when BM-MSCs were depleted from cell culture immediately before PBMCs’ stimulation (Figure 7).

## 4. Discussion

Though Th1 and Th17 cells constitute the main cell subsets implicated in the chronic inflammation of the joints observed in RA patients, other immune cells (namely monocytes/macrophages) have been increasingly recognized to play an active role in RA pathophysiology.

Lately, much attention has been paid to PB monocyte subpopulations, becoming clear they possess important functional differences, namely in what concerns to cytokine and chemokine expression profile, phagocytic activity, patrolling behavior in vivo, ability to stimulate and influence T cells’ polarization, propensity to migrate into normal or inflamed tissues, and ability to undergo differentiation into macrophages, dendritic cells, and osteoclasts [23,24,25,26,27,28]. With this background, it becomes evident the importance of studying the effect of BM-MSCs on each individual PB monocyte subset.

All these monocyte subpopulations are capable of infiltrating the synovium in RA patients [29,30], wherein they play an active role in joint damage [23]. In fact, the direct influence of monocytes on RA bone erosion became evident with a recent study demonstrating that synovial fluid mononuclear cells from RA patients spontaneously undergo differentiation into functional osteoclasts in vitro, without any additional preparation or stimulation [31]; accordingly, the number of macrophages in the synovium has been reported to correlate with joint damage [32]. Interestingly, pathological conditions are likely to modify the functional abilities of monocyte subpopulations, being reported that, in patients with psoriatic arthritis, the CD16^+^ monocytes are those with the highest ability to undergo differentiation into osteoclasts, whereas in healthy individuals the classical monocytes are the ones showing more propensity to do it [26].

The altered distribution of monocyte subsets described in RA patients further supports the involvement of these cells in the pathophysiology of the disease. In fact, the proportion of CD14^+^CD16^+^ intermediate monocytes is increased in RA synovial fluid [29] and at peripheral level [27,29,33,34]; it is also correlated to disease activity [33], and predicts the response to methotrexate therapy [23,35]. In turn, the absolute number of circulating classical and intermediate monocytes is increased in treatment-refractory patients [23]. Likewise, an increased proportion of intermediate monocytes is found among non-responder patients [23,35] and is positively correlated to disease activity [33].

The increasing evidence of the contribution of monocytes to RA pathophysiology, and the specific role of each monocyte subpopulation, highlight the relevance of our results demonstrating the ability of BM-MSCs to inhibit transversally all monocyte subsets from RA patients, ex vivo. Here, we found a consistent BM-MSC-derived suppression of TNF-α and MIP-1β (CCL4) production at protein level, by circulating monocytes. Notably, the presence of TNF-α in RA patients’ synovium leads T cells, as well as synoviocytes, to promote osteoclast activation and maturation, which can be in the basis of bone erosion [1,4]. This is further supported by studies demonstrating that anti-TNF drugs can slow or even prevent the progression of cartilage and bone damage in patients with RA [4,8]. In turn, the inhibition of MIP-1β production, mediated by BM-MSCs, was more pronounced in non-classical monocytes, a cell population composed by patrolling monocytes, in both human and mice [25,26,36], which actively migrates into injured joints and initiate joint inflammation in a murine model of RA [36]. BM-MSCs suppression of MIP-1β production by classical and non-classical monocytes, and mDCs, is a relevant finding because these cell populations can infiltrate into RA synovium [29,37] and attract Th1 cells. In fact, MIP-1β attracts CCR5^+^ cells [38], and Th1 cells infiltrating RA synovium do present CCR5 and CXCR3 at their surface [4]; intermediate monocytes also express CCR5 [27]. Thus, according to our results, MSC-based therapies may potentially hamper the recruitment of monocytes and Th1 cells into the RA inflamed joints, reduce osteoclast activity, and prevent bone destruction.

Though little information is available concerning the role of mDCs in RA, an increased percentage of mDCs [39] and upregulated levels of CD86 activation marker and CCR7 chemokine receptor [40], in the PB of RA patients have been reported; also, mDCs infiltrating the synovium express the proinflammatory cytokines IL-12p70, IL-15, IL-18, IL-23, and IFN-β [37]. Our results demonstrate that BM-MSCs can inhibit the proinflammatory cytokines production by mDCs and, at the same time, downregulate the surface levels of CD83 maturation marker and CCR7, in mDCs. Both CD83 and CCR7 are upregulated upon DC maturation and, while CCR7 is essential for DCs migration to lymph nodes, where they stimulate T cells, CD83 is implicated in the upregulation of both MHC class II and CD86 costimulatory molecule. Therefore, adequate levels of CD83 and CCR7 are essential for the development of an adaptive immune response [41,42]. Accordingly, our results show BM-MSCs can potentially restrain the deleterious effect of mDCs in the RA context.

Monocytes and DCs are important players in the activation of the adaptive immune system in RA: synovial monocytes from RA patients induce autologous peripheral CD4^+^ memory T cell polarization into Th1 and Th17cells, in vitro [29], an effect shared with RA PB monocyte-derived DCs [39]. Besides, the percentage of PB intermediate monocytes correlates positively with the percentage of Th17 cells in RA [27]; and this latter, in turn, correlates positively to DAS28 [4,43].

Our study demonstrates that BM-MSCs hinder the production of the proinflammatory cytokines implicated in the RA pathophysiology. TNF-α in the RA synovium is mainly produced by infiltrating macrophages and has the ability to activate endothelial cells and upregulate chemokine expression, promoting the infiltration of immune cells into the synovium. TNF-α is also responsible for amplifying the proinflammatory response, activating fibroblasts, chondrocytes and osteoclasts, with subsequent upregulation of matrix metalloproteinases (MMPs) expression, ultimately leading to the destruction of cartilage and bone [6,44]. Notably, by upregulating monocyte expression of IL-1β and IL-6, TNF-α promotes Th17 cell polarization [45]. All these cytokines, together with IFNγ, are detected in the synovium from RA patients and their role in cartilage destruction and bone resorption is well established [1,4,44,46,47,48]. In fact, biological drugs that block TNF-α and IL-6 undoubtedly demonstrate the critical role these cytokines play in RA. TNF-α inhibitors decrease leukocyte infiltration into the joint, reduce joint swelling, synovial vascularity, and DAS28 [3,6,49]. Furthermore, both anti-TNF and anti-IL-6 biologics stop joint damage progression in RA patients [8,48].

Remarkably, BM-MSCs possess the capability to inhibit TNF-α production (reported here) and, simultaneously, impair IL-6 expression and Th1 and Th17 cells’ function in healthy individuals [20] and RA patients [21].

Our work demonstrates BM-MSCs possess the ability to regulate monocyte and mDC’s function at different levels. And each of these levels can impact the adaptive immune system, especially the activity of Th1 and Th17 cells. BM-MSCs impair the production of proinflammatory cytokines by monocytes and mDCs; this may compromise the capability of these antigen-presenting cells to induce Th1/Th17 polarization. BM-MSCs inhibit the upregulation of CCR7, a chemokine receptor essential for cell migration to lymph nodes; this may hinder antigen-presentation. Finally, BM-MSCs downregulate chemokine expression by monocytes and mDCs, which can hamper the infiltration of immune cells into inflamed synovium, namely Th1 cells.

It has been demonstrated that IL-10 [50,51] and PGE2 [52,53], produced by MSCs, mediate the inhibition of DCs’ maturation and production of proinflammatory cytokines. In turn, the monocytes’ proinflammatory function can be hampered by CD200, expressed on MSC surface [54].

At the present moment, there are 16 phase I, II and/or phase III clinical trials, registered at www.clinicaltrials.gov (accessed on 6 February 2022), investigating MSCs-based cell therapies for RA, five of them are completed, but none of them has the results publicly available on clinicaltrials.gov. Despite that, there are several studies in small cohorts of patients reporting the safety and clinical benefits of MSCs administration to RA patients, namely in visual analog scale (VAS 100 mm) pain score, erythrocyte sedimentation rate (ESR), and 28-joint disease activity score (DAS-28) [9,55,56]. Besides, reduction in the levels of proinflammatory cytokines and Th17 cells was also observed, accompanied by an increase in Treg cells [9,57]. Clinical studies using MSCs-based cell therapy for RA patients were recently reviewed by Lopez-Santalla and colleagues [58]. RA patients’ immune cells exhibit a proinflammatory phenotype, consequently increased basal levels of TNF-α and MIP-1β in monocytes and mDCs, as well as other proinflammatory cytokines in T cells, are observed [21]. According to the current knowledge, this inflammatory environment improves the immunoregulatory function of MSCs [59], explaining the stronger immunomodulatory effect of BM-MSCs over immune cells from RA patients, in comparison to HG, as observed in the present work.

## 5. Conclusions

Recent research in RA patients and animal models pointed monocytes/macrophages as active players in RA pathophysiology. These studies have confirmed that allogeneic BM-MSCs can suppress the proinflammatory function of monocytes and mDCs, particularly to hamper TNF-α production. Our previous work, carried out in this same set of RA patients and at the same time, demonstrated allogeneic BM-MSCs also impair Th1 and Th17 cells, among others CD4^+^ and CD8^+^ T cell subsets, inhibiting the production of proinflammatory cytokines, while increasing IL-10 and TGF-β mRNA expression [21]. Altogether, our results suggest that BM-MSCs suppress the inflammatory response in RA at different levels, as they are able to hamper simultaneously antigen-presenting cells’ and T cells’ immune functions. Our findings, together with the promising results obtained after MSCs administration to RA animal models, reinforces the assumption that MSC-based therapies can be a valuable approach for RA treatment, especially for non-responder patients.

## Figures and Tables

**Figure 1 pharmaceutics-14-00404-f001:**
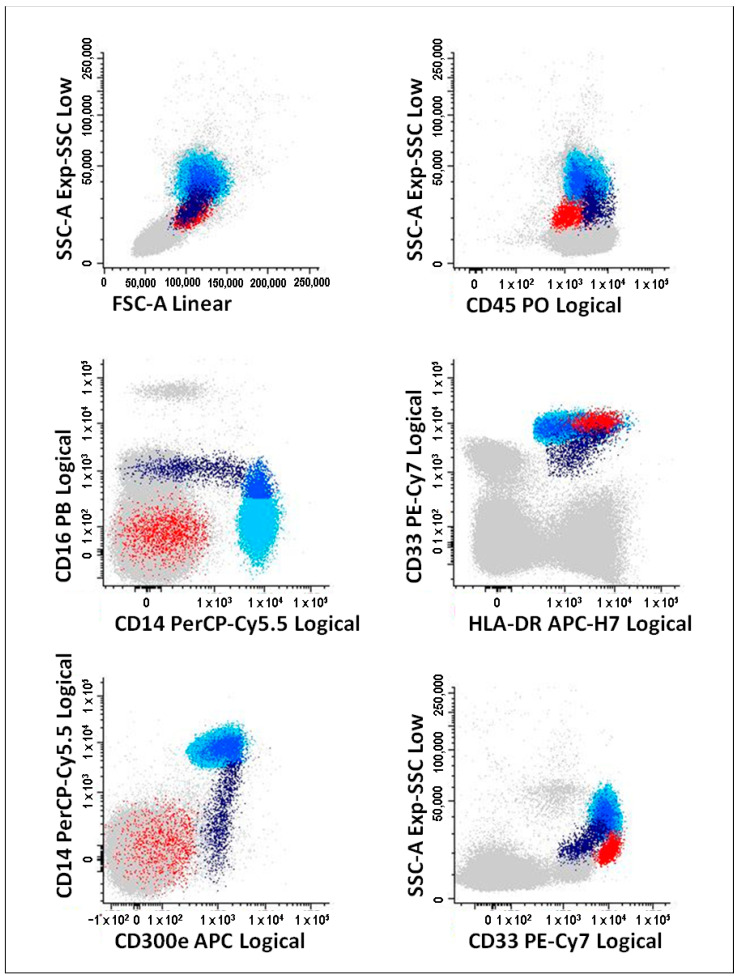
Gating strategy to identify peripheral blood monocyte subsets (classical, intermediate and non-classical) and mDCs. Classical monocytes (light blue events) were identified as CD14^+^CD16^−^, with high levels of CD33, CD300e (or IREM-2), CD45 and HLA-DR; the blue events correspond to intermediate monocytes which are CD14^+^CD16^−/+^, with high reactivity for CD300e; non-classical monocytes (dark blue events) were identified as CD14^+/−^CD16^+^, with high levels of CD45 and CD300e, and low CD33 levels; mDCs (red events) are phenotypically characterized as CD14^-^CD16^-^CD300e^-^, with low levels of CD45 and low SSC light dispersion properties, presenting higher levels of CD33 and HLA-DR than monocytes. Grey events correspond to the remaining PBMC populations: lymphocytes, plasmacytoid dendritic cells and basophils. mDCs, myeloid dendritic cells; PBMCs, peripheral blood mononuclear cells.

**Figure 2 pharmaceutics-14-00404-f002:**
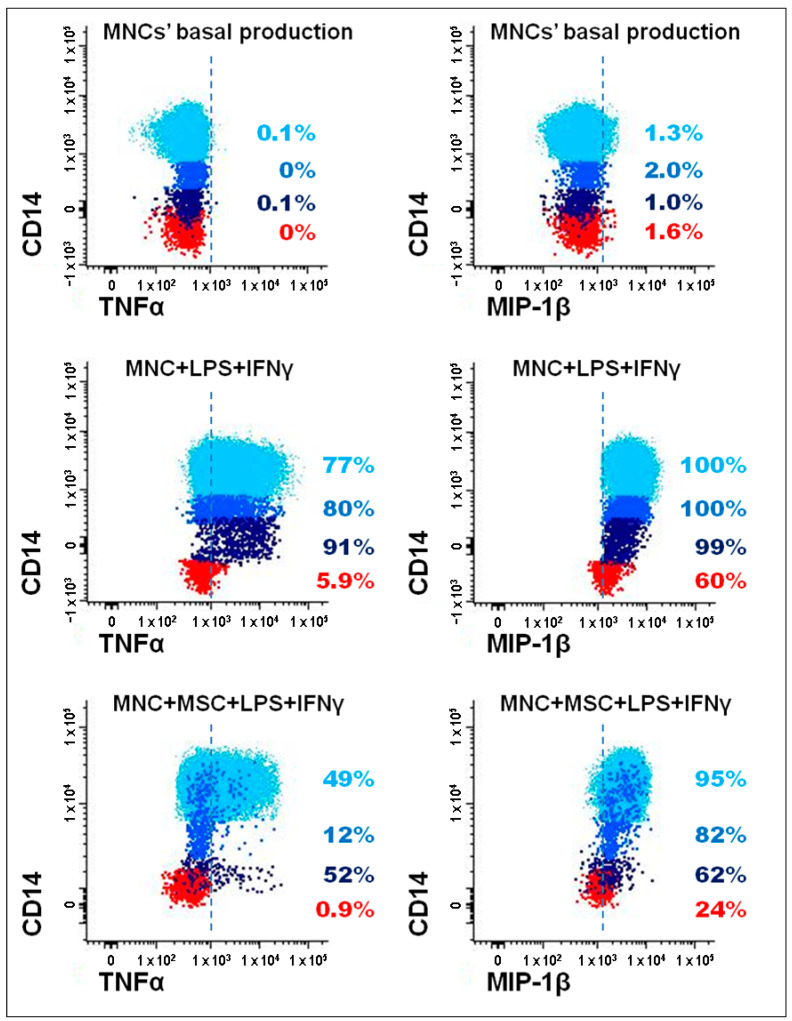
Identification of monocyte subpopulations and mDCs producing TNF-α and MIP-1β. Bivariate dot plot histograms depicting the gating strategy to identify TNF-α and MIP-1β producing cells, among monocyte subpopulations and mDCs, from a healthy individual included in the control group. Classical monocytes correspond to light blue events, intermediate monocytes are represented in blue, dark blue events correspond to non-classical monocytes, and mDCs are represented as red events. TNF-α and MIP-1β protein levels quantification is represented for three different culture conditions: unstimulated PBMCs (MNCs’ basal production), PBMCs stimulated with LPS and IFNγ (MNC + LPS+ IFNγ), and PBMCs stimulated with LPS and IFNγ in co-culture with MSCs (MNC + MSC + LPS IFNγ). IFNγ, interferon γ; LPS, lipopolysaccharide; mDCs, myeloid dendritic cells; MIP-1β, macrophage inflammatory protein-1β; MSCs, mesenchymal stromal/stem cells; PBMCs, peripheral blood mononuclear cells; TNF-α, tumor necrosis factor α.

**Figure 3 pharmaceutics-14-00404-f003:**
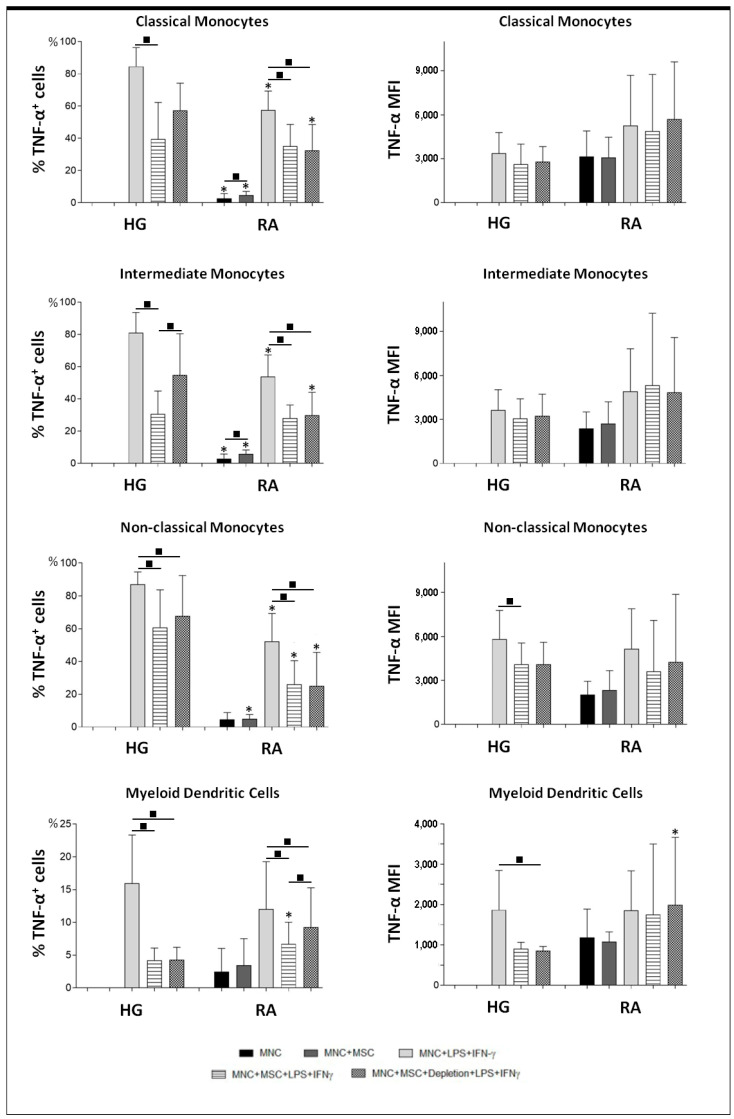
TNF-α protein levels in monocytes and mDCs from RA patients and healthy individuals. Percentage (mean ± standard deviation) of cells producing TNF-α, among monocyte subpopulations (classical, intermediate and non-classical) and mDCs, ; amount of protein (MFI) produced per cell (mean ± standard deviation), measured in the following experimental conditions: unstimulated PBMCs (MNC), non-stimulated PBMCs in co-culture with MSCs (MNC + MSC), PBMCs stimulated with LPS plus IFNγ (MNC + LPS + IFNγ), PBMCs in co-culture with MSCs and stimulated with LPS plus IFNγ in the presence of MSCs (MNC + MSC + LPS + IFNγ), PBMCs in co-culture with MSCs and stimulated with LPS plus IFNγ immediately after the depletion of MSCs from the co-culture (MNC + MSC + Depletion + LPS + IFNγ). *p* values of less than 0.05 were considered as statistically significant for Mann-Whitney test: * vs. HG, in the same culture conditions; and for Friedman’s paired-sample test: ■ between the groups indicated in the graph. HG, healthy group; IFNγ, interferon γ; LPS, lipopolysaccharide; mDCs, myeloid dendritic cells; MFI, mean fluorescence intensity; MSCs, mesenchymal stromal/stem cells; PBMCs, peripheral blood mononuclear cells; RA, rheumatoid arthritis; TNF-α, tumor necrosis factor α.

**Figure 4 pharmaceutics-14-00404-f004:**
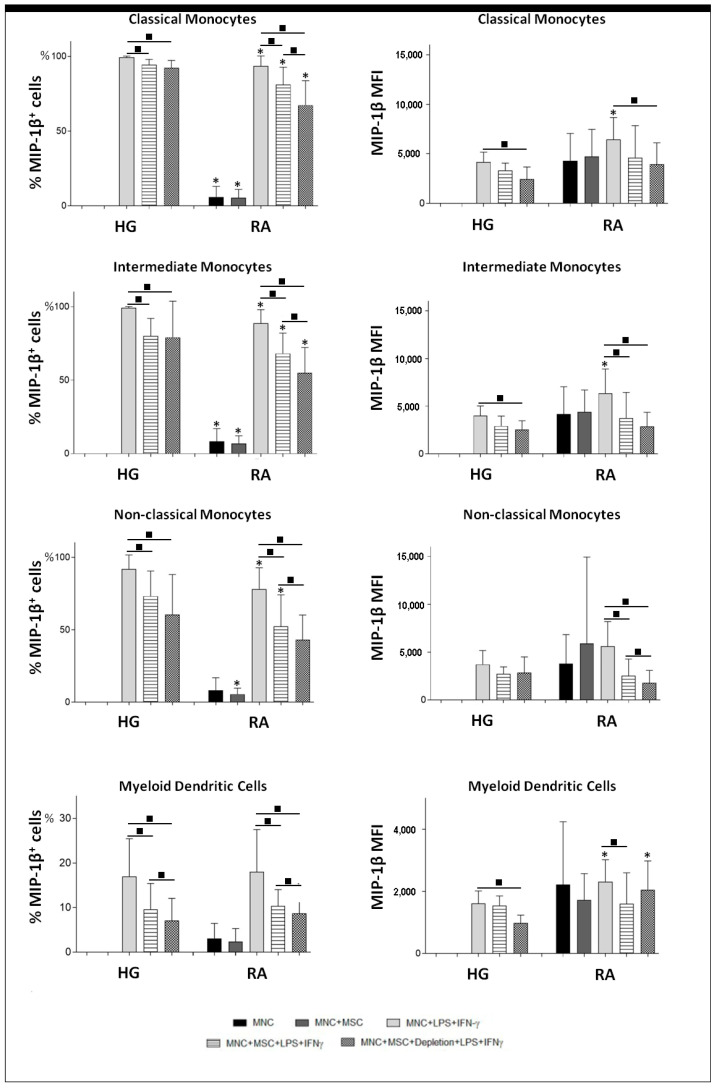
MIP-1β protein levels in monocytes and mDCs from RA patients and healthy individuals. Percentage (mean ± standard deviation) of cells producing MIP-1β, among monocyte subpopulations (classical, intermediate and non-classical) and mDCs; amount of protein (MFI) produced per cell (mean ± standard deviation), measured in the following experimental conditions: unstimulated PBMCs (MNC), non-stimulated PBMCs in co-culture with MSCs (MNC + BM-MSC), PBMCs stimulated with LPS plus IFNγ (MNC + LPS + IFNγ), PBMCs in co-culture with MSCs and stimulated with LPS plus IFNγ in the presence of MSCs (MNC + MSC + LPS + IFNγ), PBMCs in co-culture with MSCs and stimulated with LPS plus IFNγ, immediately after the MSCs were depleted from the cell culture (MNC + MSC + Depletion + LPS + IFNγ). Differences were considered statistically significant for *p* < 0.05 for Mann-Whitney test: * vs. HG, in the same culture conditions; and for Friedman’s paired-sample test: ■ between the groups indicated in the graph. HG, healthy group; IFNγ, interferon γ; LPS, lipopolysaccharide; mDCs, myeloid dendritic cells; MFI, mean fluorescence intensity; MIP-1β, macrophage inflammatory protein-1β; MSCs, mesenchymal stromal/stem cells; PBMCs, peripheral blood mononuclear cells; RA, rheumatoid arthritis.

**Figure 5 pharmaceutics-14-00404-f005:**
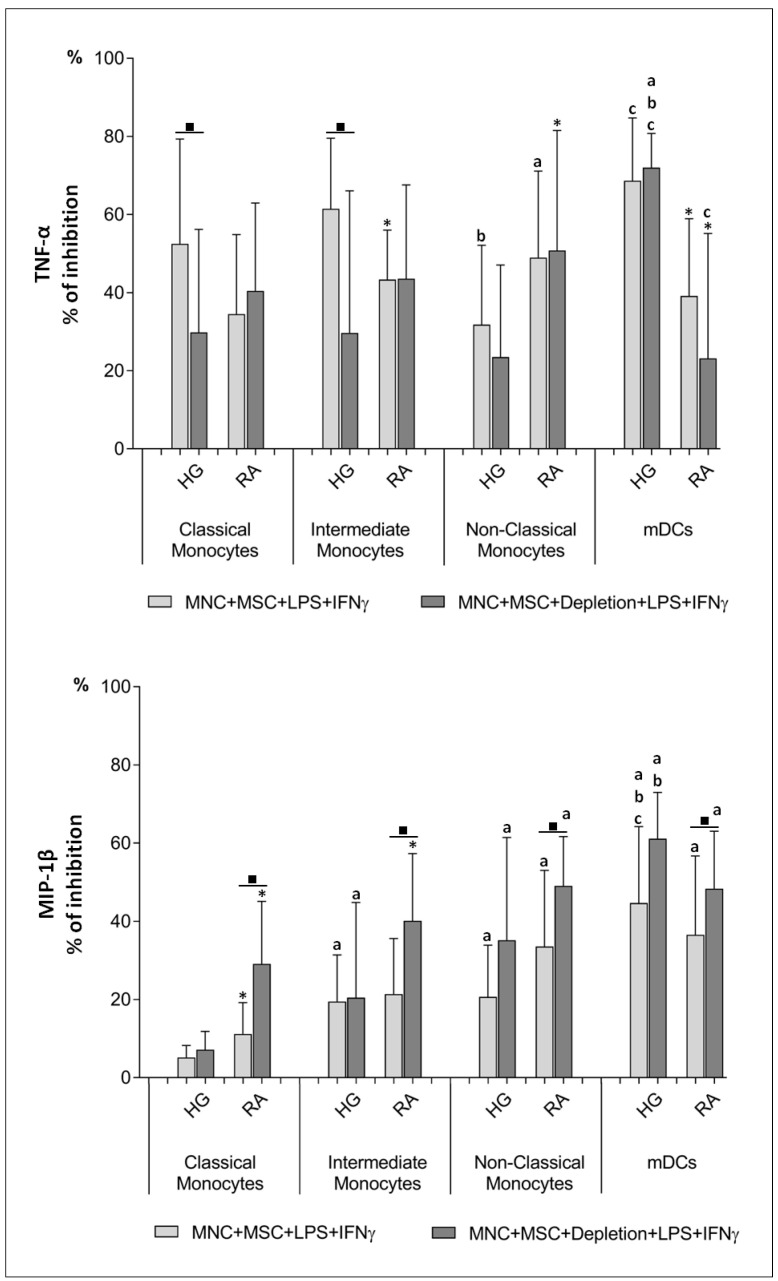
Percentage of BM-MSC-derived inhibition of TNF-α and MIP-1β protein production, by monocytes and mDCs, from RA patients and healthy individuals. Percentage of the inhibition (mean ± standard deviation), mediated by BM-MSCs, on the percentage of TNF-α- and MIP-1β-producing monocytes and mDCs, evaluated either when BM-MSCs were in the cell culture during PBMCs’ activation with LPS and IFNγ, or when BM-MSCs were removed from the cell culture immediately before PBMCs’ activation. *p* values of less than 0.05 were considered as statistically significant for Mann-Whitney test: * vs. HG, in the same cell population and the same culture condition; and for Wilcoxon paired-sample test: ■ between the groups indicated in the graph; (a) vs. classical monocytes, in the same culture condition, and within the same group of individuals; (b) vs. intermediate monocytes, in the same culture condition and within the same group of individuals; (c) vs. non-classical monocytes, in the same culture condition and within the same group of individuals. Percentage of inhibition was calculated as follows, for TNF-α: [[(percentage of TNF-α^+^ cells in PBMC + LPS + IFNγ) − (percentage of TNF-α^+^ cells in PBMC + MSC + LPS + IFNγ)]/(percentage of TNF-α^+^ cells in PBMC + LPS + IFNγ)] * 100. The same formula was applied to calculate the percentage of inhibition for MIP-1β. Accordingly, a percentage of inhibition of 100% corresponds to a complete suppression of the cytokine production, whereas a percentage of inhibition of 0% means that BM-MSCs had no effect on cytokine production by immune cells. HG, healthy group; IFNγ, interferon γ; LPS, lipopolysaccharide; mDCs, myeloid dendritic cells; MIP-1β, macrophage inflammatory protein-1β; MSCs, mesenchymal stromal/stem cells; PBMCs, peripheral blood mononuclear cells; RA, rheumatoid arthritis; TNF-α, tumor necrosis factor-α.

**Figure 6 pharmaceutics-14-00404-f006:**
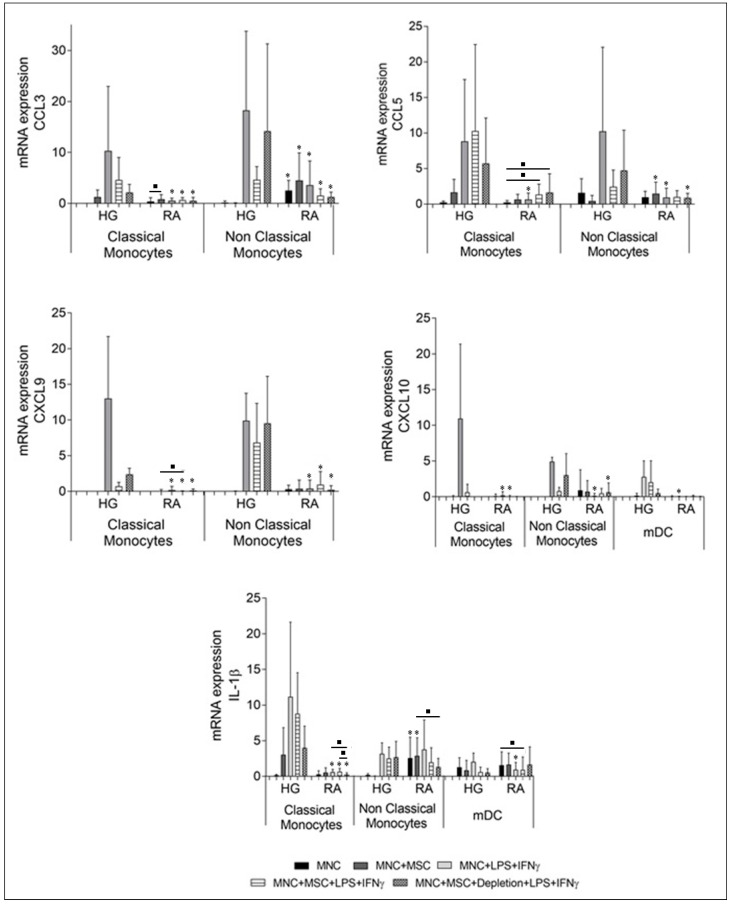
Effects of BM-MSCs over cytokines and chemokines mRNA expression by monocytes and mDCs from healthy individuals and RA patients. Semi-quantitative determination of CCL3, CCL5, CXCL9, CXCL10 and IL-1β mRNA expression in FACS-purified classical monocytes and non-classical monocytes, and of CXCL10 and IL-1β in mDCs. mRNA levels were measured under the following culture conditions: non-stimulated PBMCs (MNC), non-stimulated PBMCs in co-culture with MSCs (MNC + BM-MSC), PBMCs stimulated with LPS plus IFNγ (MNC + LPS + IFNγ), PBMCs in co-culture with MSCs and stimulated with LPS plus IFNγ in the presence of MSCs (MNC + MSC + LPS + IFNγ), PBMCs in co-culture with MSCs and stimulated with LPS plus IFNγ immediately after the depletion of MSCs from the culture system (MNC + MSC + Depletion + LPS + IFNγ). The normalized expression levels of the genes were calculated by using the delta-Ct method. Statistically significant differences were considered for *p* < 0.05 for Mann-Whitney test: * vs. HG, in the same culture condition and in the same cell population; and for Friedman’s paired-sample test: ■ between the groups indicated in the graph. FACS, fluorescence-activated cell sorting; HG, healthy group; IFNγ, interferon γ; IL, interleukin; LPS, lipopolysaccharide; mDCs, myeloid dendritic cells; MSCs, mesenchymal stromal/stem cells; PBMCs, peripheral blood mononuclear cells; RA, rheumatoid arthritis.

**Figure 7 pharmaceutics-14-00404-f007:**
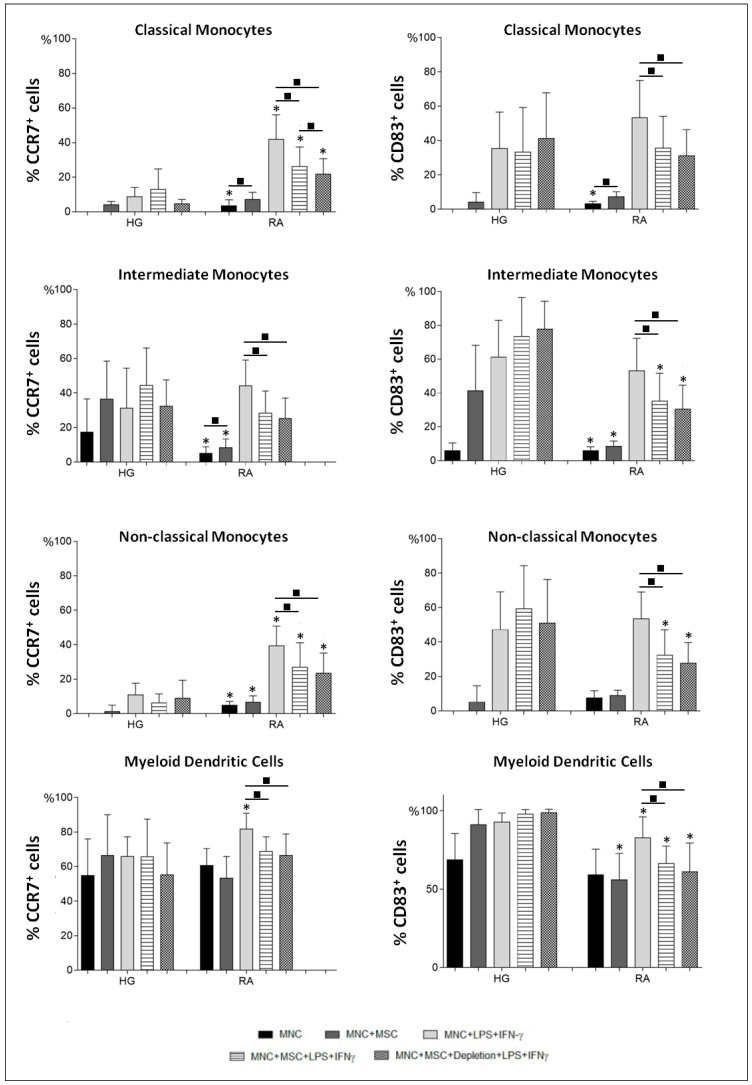
Influence of BM-MSCs over the protein levels of CCR7 and CD83 by monocyte subsets and mDCs from RA patients and healthy individuals. Percentage (mean ± standard deviation) of CCR7^+^ and CD83^+^ cells, among monocyte subpopulations (classical, intermediate and non-classical) and mDCs measured in the following experimental conditions: unstimulated PBMCs (MNC), non-stimulated PBMCs in co-culture with BM-MSCs (MNC + MSC), PBMCs stimulated with LPS plus IFNγ (MNC + LPS + IFNγ), PBMCs in co-culture with BM-MSCs and stimulated with LPS plus IFNγ in the presence of MSCs (MNC + MSC + LPS + IFNγ), PBMCs in co-culture with BM-MSCs and stimulated with LPS plus IFNγ immediately after the depletion of BM-MSCs from the cell culture (MNC + MSC + Depletion + LPS + IFNγ). *p* values of less than 0.05 were regarded as statistically significant for Mann-Whitney test: * vs. HG, in the same culture condition; and for Friedman’s paired-sample test: ■ between the groups indicated in the graph. HG, healthy group; IFNγ, interferon γ; LPS, lipopolysaccharide; mDCs, myeloid dendritic cells; MSCs, mesenchymal stromal/stem cells; PBMCs, peripheral blood mononuclear cells; RA, rheumatoid arthritis.

**Figure 8 pharmaceutics-14-00404-f008:**
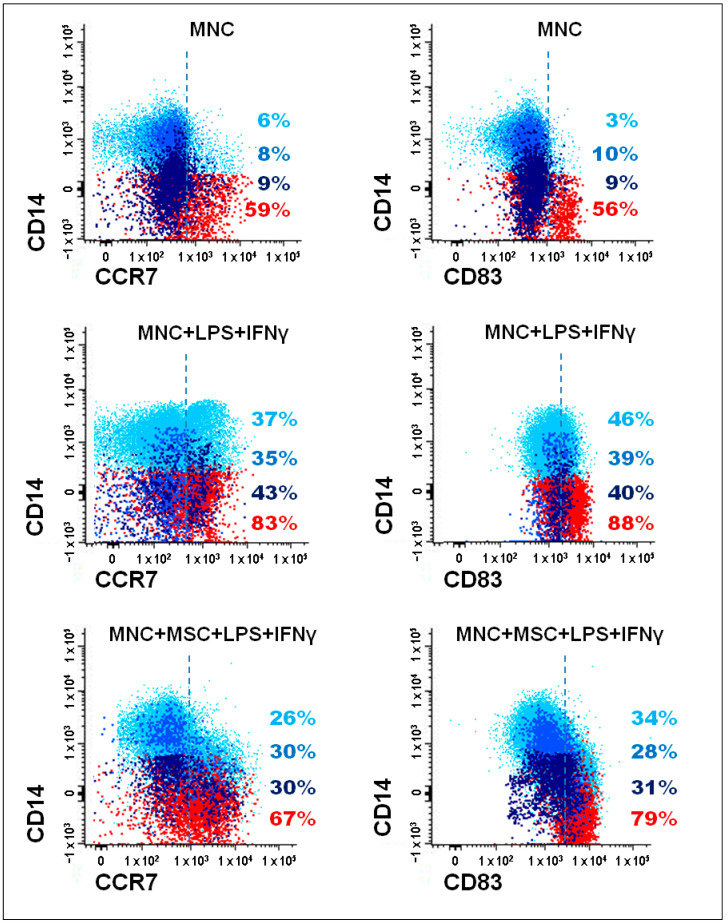
Percentage of CCR7^+^ and CD83^+^ cells, among classical, intermediate, and non-classical monocytes and mDCs. Bivariate dot plot histograms depicting the gating strategy to identify CCR7^+^ and CD83^+^ cells, among monocyte subpopulations and mDCs, from a RA patient. Classical monocytes correspond to light blue events, intermediate monocytes are represented in blue, dark blue events correspond to non-classical monocytes, and mDCs are represented as red events. The percentage of CCR7^+^ and CD83^+^ cells was evaluated for the three different culture conditions: unstimulated PBMCs (MNC), PBMCs stimulated with LPS and IFNγ (MNC + LPS+ IFNγ), and PBMCs stimulated with LPS and IFNγ in co-culture with MSCs (MNC + MSC + LPS+ IFNγ). IFNγ, interferon γ; LPS, lipopolysaccharide; mDCs, myeloid dendritic cells; MSCs, mesenchymal stromal/stem cells; PBMCs, peripheral blood mononuclear cells.

**Table 1 pharmaceutics-14-00404-t001:** Panel of monoclonal antibodies used for immune cells’ phenotypic characterization, indicating the commercial source and clone.

Fluorochromes								
Tubes	PB	PO	FITC	PE	PerCP-Cy5.5	PE-Cy7	APC	APC-H7
**1**	**CD16** BD Pharmingen 3G8	**CD45** Invitrogen HI30	**CD83** Beckman Coulter HB15a	**CCR7** BD Pharmingen 3D12	**CD14** BD Pharmingen M5E2	**CD33** Beckman Coulter D3HL60.251	**CD300e** Immunostep SL UP-H2	**HLA-DR** BD L243
**2**	**CD16** BD Pharmingen 3G8	**CD45** Invitrogen HI30	**cyTNF-α** BD Pharmingen MP6-XT22	**cyMIP-1β** BD Pharmingen D21-1351	**CD14** BD Pharmingen M5E2	**CD33** Beckman Coulter D3HL60.251	**CD300e** Immunostep SL UP-H2	**HLA-DR** BD L243
**3**	**CD16** BD Pharmingen 3G8			**CD123** Beckman Coulter SSDCL Y107D2	**CD14** BD Pharmingen M5E2	**CD33** Beckman Coulter D3HL60.251	**CD300e** Immunostep SL UP-H2	**HLA-DR** BD L243

Abbreviations: APC, allophycocyanin; APC-H7, allophycocyanin-hilite 7; FITC, fluorescein isothiocyanate; PB, pacific blue; PE, phycoerythrin; PE-Cy7, phycoerythrin-cyanine 7; PerCP-Cy5.5, peridinin chlorophyll protein-cyanine 5.5; PO, pacific orange. Commercial sources: BD (Becton Dickinson Biosciences, San Jose, CA, USA); BD Pharmingen (San Diego, CA, USA); Beckman Coulter (Miami, FL, USA); Immunostep S.L (Salamanca, Spain); Invitrogen, Life Technologies (Carlsbad, CA, USA).

## Data Availability

The data presented in this study are available on request from the corresponding author. The data are not publicly available due to applicable ethical or privacy regulations.

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
