# Peer review of "Human Bone Marrow Mesenchymal Stromal/Stem Cells Regulate the Proinflammatory Response of Monocytes and Myeloid Dendritic Cells from Patients with Rheumatoid Arthritis"

_pharmaceutics, 2022, doi:10.3390/pharmaceutics14020404_

Round 1

Reviewer 1 Report

The manuscript is dedicated to investigate how human bone marrow mesenchymal stromal/stem cells regulate the proinflammatory response of monocytes and myeloid dendritic cells that were isolated from patients with rheumatoid arthritis. The manuscript needs more clear data explanation and presentation. There are some pending questions.

  1. The patients’ selection rises some questions since women and men are mixed in selected groups and even with different proportions in the groups that could negatively influence the final results.
  2. It is not clear, have the authors used autologous BM-MSCs or allogeneic?
  3. If the BM-MSCs, after co-culture, were separated according to the CD271 marker, the BM-MSCs population before co-culture also should be separated according the CD271 level. Such information is missing. The CD271 is one of the BM-MSC markers but it does not mean that 100% of used BM-MSCs population express CD271.
  4. 1. The more than one graph in the Fig. should have letter numbering or titles of the graphs in order to better understand what are the authors talking about. The y axis writing is very difficult to read and follow the results.
  5. 1. The x and y axes of the graphs are more than strange – “….logical”, when the axes are clearly logarithmic.
  6. The interpretation of Fig. 3 - 5 should be separated and better explained. The text should clearly state are these cytokines secreted of measured on monocytes. Since it is flow cytometry measurements, it is possible to assume that everything is on the monocyte or DC surface.

             The flow cytometry data are usually shown as either cell population having bound fluorescence antibody or by the mean of fluorescence intensity (MFI) of bound antibody. Both measurements give similar results, but MFI is more sensitive. In this study, the cell population measurement somehow is named as “…frequency”, which is not clear terminology and in parallel the MFI measurement data were also added. The Fig. 3 and 4 are overloaded with MFI measurement since Fig 5 is summarizing only cell population data. The y axis of Fig.5 is again wrongly named – “% inhibition of TNF or MIP-1beta” since not all data show the inhibition. Why not to simply name as a level of TNF alpha or MIP-1beta.

  1. Usually it is agreed that “expression” is used for the genes, while for the protein – the level of proteins on the cells, in the cells or secreted. In this study it is difficult to follow are these cytokines secreted (like TNFalpha), or measured their protein levels in some other level. If there were no ELISA measurements, it is not clear how secreted TNF-alpha can be analyzed by flow cytometer.
  2. Methodological part as well as the result part should be more clearly described and explained.
  3. 6. The only CXCL10 and IL-1beta were detected in mDC. It is not clear whether these genes have not been investigated or not established in mDC.
  4. 7. The same remarks concerning “….frequency” on the y axis. It should be a percent of population positive for the biomarker. Would be more clear to follow results if graphs would have titles or label by letters.
  5. Some sentences should be also checked: line 452 “In turn, BM-MSC-derived inhibition of MIP-1β….”, line 468 “mDCs and downregulate their surface expression CD83 maturation marker and CCR7“.

Reviewer 2 Report

I have read with interest article entitled: “Human bone marrow mesenchymal stromal/ stem cells regulates the pro-inflammatory response of monocyte and myeloid dendritic cells from patients with rheumatoid arthritis”. In order to improve manuscript quality some changes have to be performed:

  1. Last paragraph from Introduction can be removed because it describe results.
  2. Why Authors decided to use MSCs from bone marrow?
  3. PBMCs were cultured in RPMI while BM-MSCs in DMEM. In co-coculture only RPMI was used, if this affected BM-MSCs properties?
  4. Allogenic BM-MSCs were used in this study, could it have had an effect on obtained results?
  5. Authors wrote that BM-MSCs can suppress the proinflammatory function of monocytes and mDCs. What is the possible mechanism of action of BM-MSCs in this process?
  6. In clinicaltrial.gov a lot of study about MSCs application in RA can be found. Authors should discuses potential of this therapy as a standard methods of RA treatment.
  7. Some small grammatical mistakes within the manuscript can be found.

Reviewer 3 Report

The authors of the study have demonstrated that human bone marrow mesenchymal stromatolites cells (MSCs) help to regulate the pro-inflammatory response of monocytes and dendritic cells from patient with rheumatoid arthritis. The data specifically show that the monocytes and dendritic cells express pro-inflammatory cytokines, although in co-culture with MSCs, IL-1ß and CCL3 have downregulated gene expression levels in this model, whilst CD83 and CCR7 positive cells were reduced. 

There remain open questions in this manuscript that need to be addressed prior to further consideration for publication.

1) In the collection of patients, there is a difference in the number of male and female donors used in the study. Did you see male vs female difference in RA patients with respect to cytokine release - show a graph to prove this point. Additionally, were the same genders used for the MSC co-culture model (i.e. female MSCs with female dendritic cells) ? The latter may influence the results.

2) The graphs are difficult to understand, as you cannot see the groups that are significant between each other. Please use a line between significant groups to clearly show the differences.

3) For the gene expression, is this relative to the housekeeper (ΔCt) or to a control group (ΔΔCt). Please clearly indicate in the graphs.

4) Images of labelled cells to show the depletion in CD83 and CCR7 expression are require to clearly illustrate the data in figure 7.

5) Do the authors have ELISA data showing a reduction in cytokine release in their co-culture model, alongside the given gene expression data. Please provide a graph to show this.

6) Do you have any data investigating pathways influenced by this co-culture model, E.g. NF-κB ? Western blot analysis of this pathway would be good to provide or a comment within the discussion.

Round 2

Reviewer 1 Report

The authors significantly improved the manuscript, added additional explanations to the method and result part. The Figures also were corrected. 

326 line. As it was remarked earlier, such terms as  "TNFα and MIP-1β protein expression" should be avoided, since it is not gene expression. It confuses the reader. It should be "TNFα and MIP-1β protein level". The last figure 8 legend title also better to write "Percent of the cells...." since it is measured protein level by flow cytometer, not gene expression. Hopefully the authors will correct the usage of these terms all over the manuscript. The article can be published.

Author Response

Dear reviewer,

To follow your recommendation, we proceeded to the substitution of “protein expression” by “protein levels” throughout all the manuscript. We also altered the Figure’s 8 legend according to your suggestion. Hopefully, we have addressed all of your concerns.

Reviewer 3 Report

The authors have addressed my concerns stated in my review appropriately.

Author Response

Dear reviewer,

Thank you for your careful revision, which definitely contributed to improve our manuscript, resulting in a more compelling paper.

This manuscript is a resubmission of an earlier submission. The following is a list of the peer review reports and author responses from that submission.